# Coexistence of Retinitis Pigmentosa and Ataxia in Patients with PHARC, PCARP, and Oliver–McFarlane Syndromes

**DOI:** 10.3390/ijms25115759

**Published:** 2024-05-25

**Authors:** Anna Wawrocka, Joanna Walczak-Sztulpa, Lukasz Kuszel, Zuzanna Niedziela-Schwartz, Anna Skorczyk-Werner, Jadwiga Bernardczyk-Meller, Maciej R. Krawczynski

**Affiliations:** 1Department of Medical Genetics, Poznan University of Medical Sciences, 60-806 Poznan, Poland; jsztulpa@ump.edu.pl (J.W.-S.); kuszel@ump.edu.pl (L.K.); askorczyk@ump.edu.pl (A.S.-W.); mrkrawcz@ump.edu.pl (M.R.K.); 2Department of Ophthalmology, University Clinical Hospital in Poznan, 60-569 Poznan, Poland; z.niedziela@gmail.com; 3Specialized Ophthalmology Office and Eye Electrophysiology Laboratory, ERETINA, 60-189 Poznan, Poland; jbmeller@poczta.onet.pl; 4Centers for Medical Genetics GENESIS, 60-406 Poznan, Poland

**Keywords:** retinitis pigmentosa and ataxia, PHARC syndrome, PCARP syndrome, Oliver–McFarlane syndrome (OMCS), diagnostic NGS panel, whole-exome sequencing

## Abstract

Retinitis pigmentosa (RP) is an inherited retinal dystrophy caused by the loss of photoreceptors and retinal pigment epithelial atrophy, leading to severe visual impairment or blindness. RP can be classified as nonsyndromic or syndromic with complex clinical phenotypes. Three unrelated Polish probands affected with retinitis pigmentosa coexisting with cerebellar ataxia were recruited for this study. Clinical heterogeneity and delayed appearance of typical disease symptoms significantly prolonged the patients’ diagnostic process. Therefore, many clinical and genetic tests have been performed in the past. Here, we provide detailed clinical and genetic analysis results of the patients. Whole-exome sequencing (WES) and targeted NGS analysis allow the identification of four novel and two previously reported variants in the following genes: *ABHD12*, *FLVCR1*, and *PNPLA6.* The use of next-generation sequencing (NGS) methods finally allowed for confirmation of the clinical diagnosis. Ultra-rare diseases such as PHARC, PCARP, and Oliver–McFarlane syndromes were diagnosed in patients, respectively. Our findings confirmed the importance of the application of next-generation sequencing methods, especially in ultra-rare genetic disorders with overlapping features.

## 1. Introduction

Retinitis pigmentosa (RP, MIM 268000) is a heterogeneous group of inherited retinal disorders affecting 1 in 4000 people worldwide. RP is characterized by night blindness, loss of peripheral vision, and reduced central vision, leading to severe visual impairment or blindness. The disorder can be classified as isolated (nonsyndromic) or syndromic with extraocular involvement [1,2]. Many syndromes are associated with RP, including more common, like Usher syndrome, and ultra-rare genetic disorders like PHARC, PCARP, or Oliver–McFarlane syndrome (OMCS). Apart from the symptoms of RP, these three autosomal recessive diseases also have ataxia in common, which becomes apparent as the disease progresses.

PHARC syndrome (MIM 612674) (polyneuropathy, hearing loss, ataxia, retinitis pigmentosa, and early-onset cataract) is a neurodegenerative disorder caused by defects in the *ABHD12* gene. *ABHD12* (MIM 613599) encodes the α/β-hydrolase domain-containing protein 12, a hydrolytic enzyme involved in endocannabinoid metabolism. PHARC is a genetically heterogeneous and clinically variable disease [3]. Symptoms are slowly progressive and begin in childhood or adolescence. Patients affected with PHARC syndrome can be misdiagnosed due to some overlapping features with other conditions, including Refsum disease and Charcot–Marie–Tooth disease [4,5]. Therefore, the differential diagnosis of PHARC syndrome from disorders with similar symptoms is essential for patients. 

A neurodegenerative syndrome characterized by posterior column ataxia and retinitis pigmentosa is called PCARP (MIM 609033). The disease begins in infancy with areflexia and symptoms of retinal degeneration. The progression is characterized by visual field restriction, night blindness, and later onset secondary ataxia [6]. PCARP is caused by pathogenic variants in the *FLVCR1* (MIM 609144) gene that encodes the feline leukemia virus subgroup C receptor 1, a heme-transporter protein that maintains the intracellular heme concentration [7,8]. 

Oliver–McFarlane syndrome (MIM 275400) is characterized by trichomegaly, severe retinitis pigmentosa, and multiple pituitary hormone deficiencies, including growth hormone (GH), gonadotropins, and thyroid-stimulating hormone (TSH) [9]. Untreated disease leads to intellectual impairment and profound short stature in affected individuals. OMCS is a highly clinically heterogeneous disease, and half of the reported patients had spinocerebellar involvement, including inter alia ataxia or spastic paraplegia [10]. Disease results from defects in the patatin-like phospholipase domain-containing protein 6 gene (*PNPLA6*, MIM 603197).

In this study, we report on detailed clinical and genetic analyses of three patients with retinitis pigmentosa coexisting with cerebellar ataxia.

## 2. Results

Three unrelated patients of Polish origin affected with ataxia and retinitis pigmentosa were recruited for this study. The clinical diagnosis was established by detailed medical history analysis and ophthalmological and neurological assessment.

### 2.1. Clinical Features of Patient 1 (P1)

The patient was a 33-year-old male. The parents did not report any abnormalities concerning psychomotor development during infancy. At the age of 6 years, he was diagnosed with color blindness and hyperopia with astigmatism. The patient reported decreasing night vision at the age of 13. Due to decreased visual acuity, central scotoma in the visual field, color blindness, and electroretinographic changes, the patient was initially diagnosed as a late form of cone-rod dystrophy. However, lack of photophobia, early onset of night vision abnormalities, unusual character of central scotoma (photopsia), and no “bull’s eye” appearance of the retina indicated an unusual course of the disease and therefore looks more like atypical RP (Figure 1). Due to the emerging symptoms of sensorimotor polyneuropathy, Charcot–Marie–Tooth disease was suspected. In addition, a diagnosis of multiple sclerosis was performed. Brain MRI allowed the exclusion of SM and it showed a pineal gland cyst. New symptoms have emerged in recent years, such as sensorineural hearing loss with tinnitus, muscle pain and weakness, and chronic fatigue. Therefore, a suspicion of mitochondrial disease was raised. Physical examination revealed signs of ataxia.

#### Molecular Results of Patient 1 (P1)

Due to suspicion of cone-rod dystrophy in the patient, the *ABCA4* gene was sequenced but did not show any pathogenic variants. Subsequently, on account of this patient’s emerging motor/sensory neuropathy symptoms, a multiplex ligation-dependent probe amplification (MLPA) analysis of the *PMP22* gene was performed. The result was normal. NGS panel sequencing associated with Charcot–Marie–Tooth neuropathy did not reveal any putative disease-causing variants in the patient. Next, mitochondrial DNA sequencing was made to screen for possible mitochondrial disorders. However, no pathogenic or possibly pathogenic variant was revealed. Finally, WES results allowed us to identify disease-causing variants in the compound heterozygous state. The novel deletion of 186,118 bp (chr20:25,319,102-25,505,220; hg38) encompassed 5′UTR, exons 1-4, of the *ABHD12*, the entire sequence of the adjacent gene *GINS1* (5′UTR, exons 1-7, and 3′UTR), and a part of the *NINL* gene (3′UTR, exons 6-24). The deletion was combined with previously reported nonsense variant c.1063C˃T, p.(Arg355Ter) in exon 12 of the *ABHD12* gene. qPCR (quantitative polymerase chain reaction) and Sanger sequencing were utilized to narrow down the genomic coordinates of the detected deletion (Appendix A). Segregation analysis of the patient’s parents’ variants was performed through a qPCR, and Sanger sequencing revealed that the mother was a carrier of the deletion and the father carried the nonsense variant (Figure 2).

### 2.2. Clinical Features of Patient 2 (P2)

The patient was a 61-year-old female, healthy in childhood and adolescence. From age 22, she observed the first balance, sensory problems, and nystagmus. Over time, severe sensorineural polyneuropathy and ataxia developed. The first symptoms of night blindness were noticed at the age of 20. Based on the results of ophthalmological examinations, RP was diagnosed. Auditory evoked potentials were normal in the patient. MRI (magnetic resonance imaging) of the head allowed the exclusion of multiple sclerosis. Moreover, normal phytanic acid and vitamin E levels ruled out Refsum’s disease and ataxia with isolated vitamin E deficiency, respectively. Electromyography examination revealed severe sensorineural polyneuropathy. The typical symptoms of RP observed in the patient, coexisting with neurodegenerative symptoms, could suggest the diagnosis of spinocerebellar ataxia or mitochondrial disease (Figure 1).

#### Molecular Results of Patient 2 (P2)

Mitochondrial NARP syndrome and spinocerebellar ataxia were considered in the patient. However, molecular analyses, including mtDNA sequencing (for NARP syndrome) and trinucleotide repeats in *ATXN* genes (for spinocerebellar ataxia) analysis, did not allow for detecting disease-causal variants. Therefore, both disorders could be excluded in the patient. Only WES enabled the identification of two novel compound heterozygous missense variants c.648C˃A, p.(Phe216Leu) and c.733A˃T, p.(Asn245Tyr) in exon 1 in the *FLVCR1* gene. WES results were verified by Sanger sequencing (Figure 2). Variants segregation analysis in the family was not possible due to the death of the patient’s parents.

### 2.3. Clinical Features of Patient 3 (P3)

Patient 3 was a 15-year-old male. The patient’s psychomotor development was delayed from infancy, and he could walk with no assistance at 20 months. He presented an abnormal ataxic walk. The patient had characteristic long and curled upwards eyelashes—trichomegaly (Figure 1). The parents noticed the first symptoms of night blindness in the second year of life. Retinitis pigmentosa was suspected at the age of 4. Ophthalmological examinations allowed the confirmation of atypical RP (Figure 1). Tandem mass spectrometry (tandem-MS) and gas chromatography–mass spectrometry (GC–MS) analyses were performed on the patient and allowed metabolic disorders to be excluded. Growth retardation was observed in the patient, and therefore endocrinological analyses were performed. The results of endocrinological tests indicated multihormonal hypopituitarism. MRI of the brain revealed a reduced pituitary gland. 

Detailed results of clinical examinations in all patients are presented in Table 1.

#### Molecular Results of Patient 3 (P3)

Targeted NGS allowed for detection in the patient: the *PNPLA6* gene variants in a compound heterozygous state: novel nonsense variant c.1387C˃T p.(Gln463Ter) in exon 12, and previously reported missense variant c.3343G˃A p.(Asp1115Asn) in exon 27. The presence of both variants was confirmed by Sanger sequencing, and segregation analysis revealed that the mother and father are each carriers of p.(Gln463Ter) or p.(Asp1115Asn), respectively (Figure 2).

None of the novel variants identified in this study were present in the HGMD^®^Professional, gnomAD SVs (v2.1) (genome Aggregation Database), ClinVar, and in the Leiden Open Variation Database (LOVD) and, therefore, based on the ACMG guidelines, none were assessed as pathogenic or likely pathogenic variants (Table 2).

## 3. Discussion

In this study, we report on three patients presenting with symptoms of retinitis pigmentosa coexisting with cerebellar ataxia. Due to the clinical heterogeneity and delayed onset of some typical disease symptoms, patients have undergone a long diagnostic path, and only next-generation sequencing analysis allowed confirmation of the clinical diagnosis. Using WES and targeted NGS analysis, we established the molecular background of the disease in all patients. Among all causal variants, four novel and two previously reported variants were identified in the following genes: *ABHD12*, *FLVCR1*, and *PNPLA6* (Table 2). 

Patient 1 was found to be a compound heterozygote for a nonsense variant c.1063C˃T, p.(Arg355Ter), combined with a 186,118 bp deletion in the *ABHD12* gene. The nonsense mutation p.Arg355Ter has already been reported in another patient with PHARC syndrome [11]. Copy number variants (CNVs) analysis using WES data allowed for the identification of a novel heterozygous deletion on the other allele in the patient. The deletion encompasses the fragment of *ABHD12* (exons 1–4), the entire sequence of the *GINS1* gene, and a large part of the *NINL* gene (exons 6–24). The heterozygous deletion was also confirmed in the patient’s asymptomatic mother. Deletion involving a fragment of the *ABHD12* gene and a portion of the *GINS1* gene was described earlier. Chen et al. reported a patient who carries a 59 kb deletion that encompasses exon 1 of *ABHD12* and exons 1–4 of the *GINS1* gene and nonsense variant (c.1129A>T; p.Lys377*) on the other allele [12]. It was the first description of compound heterozygosity in a patient with PHARC syndrome. Furthermore, a description of a small homozygous deletion of 14 kb covering the 5′UTR and exon 1 of the *ABHD12* gene in a patient with PHARC was published in 2010 [3]. To our best knowledge, the deletion detected in the proband in this study is the first aberration with such size and content. *NINL* gene deletion has never been reported in PHARC patients before. NINL is a centrosomal protein, a physical and genetic interaction partner of CC2D2A; both localized at the base of cilia. Research has shown that ninl or cc2d2a knockdown in zebrafish causes a similar retinal phenotype, including photoreceptor outer segment loss, accumulation of vesicles, and mislocalization of opsins [13,14]. 

Clinical presentation in patients with PHARC syndrome with regard to the disease’s first symptoms, severity, and its progression can vary widely. Polyneuropathy is usually one of the first reported findings, manifested generally in childhood [3,11,15]. Other studies point to hearing loss as the first symptom observed in PHARC patients. Ocular involvement is usually noticeable later as the disease progresses [5]. However, in patient 1, the ophthalmological symptoms of decreased night vision and reduced visual acuity appeared first. In contrast, polyneuropathy and hearing loss were first observed in P1 in the second decade of life. Analysis of approximately 50 reported PHARC patients so far did not allow for determining the evident correlation between the type and position of *ABHD12* pathogenic variants as well as the severity and progression of the disease.

Patient 2 presented with typical features of PCARP syndrome. However, the onset of retinal degeneration symptoms was visible quite late (at 20 years), while in most patients they were present in the first decade of life. WES allowed the revelation of two novel, missense variants in a compound heterozygous state (c.648C˃A, p.(Phe216Leu) and c.733A˃T, p.(Asn245Tyr)) in the *FLVCR1* gene. Both affected amino acids are localized in the transmembrane domain (TMD) of the FLVCR1 protein. Previous studies have suggested that TMD variants lead to misfolding in the endoplasmic reticulum and FLVCR1 protein degradation in the lysosomes, and, therefore, a loss of protein’s heme export activity [16]. In vitro studies showed that disruption of the FLVCR1 heme exporter leads to increased reactive oxygen species and intracellular heme accumulation, contributing to the dysfunction of several organs and tissues, such as the brain, retina, and spinal cord [7,17]. Due to the very high expression of the gene in the retina and slightly lower in posterior columns, the main symptoms of the disease are located primarily in these organs [18]. 

The *PNPLA6* gene is implicated in several neurodegenerative conditions, including Gordon–Holmes syndrome, Boucher–Meuhäuser, spastic paraplegia type 39, Laurence–Moon, and Oliver–McFarlane syndrome [10,19]. Oliver–McFarlane and Laurence–Moon syndromes share similar phenotypes, including severe retinal degeneration, pituitary dysfunction with short stature, early onset of ataxia, and peripheral neuropathy. Trichomegaly, observed in Oliver–McFarlane patients, is the only congenital feature differentiating these two syndromes [10]. Patient 3 presents typical OMCS symptoms of the disease. At the age of 7, he was diagnosed clinically with Oliver–McFarlane syndrome based on the phenotype of retinitis pigmentosa, trichomegaly, multihormonal hypopituitarism, growth retardation, and ataxia (Figure 1). The current targeted NGS analysis confirmed the clinical diagnosis. Two variants in a compound heterozygous state were identified; one novel c.1387C>T p.(Gln463Ter) and previously described, c.3343G>A p.(Asp1115Asn) in the *PNPLA6* gene. Variant p.(Asp1115Asn) is located in the NTE protein’s patatin-like catalytic (NEST) domain. 

In addition to the abovementioned disorders, ataxia and RP may also appear in the course of other disorders, such as NARP syndrome (MIM 551500), Refsum disease (MIM 266500), Laurence–Moon syndrome (MIM 245800), abetalipoproteinemia (MIM 200100), spinocerebellar ataxia type 7 (MIM 164500), and congenital disorder of glycosylation type 1A (MIM 212065). Refsum disease (ARD) is a rare peroxisomal disorder caused by variants in the *PHYH* gene encoding phytanoyl-CoA hydroxylase. The main symptoms are retinitis pigmentosa, polyneuropathy, and cerebellar ataxia. Additionally, patients may present hearing loss, anosmia, ichthyosis, cataract, cardiac arrhythmia, and elevated protein level in cerebrospinal fluid [20]. Due to the symptoms manifested in P2, Refsum’s disease was suspected; however, a normal level of phytanic acid allowed the exclusion of the disease. NARP is a syndromic mitochondrial disorder characterized mainly by proximal muscle weakness, axonal neuropathy, RP, and cerebellar ataxia. Disease results from pathogenic variants in the *MT-ATP6* gene that encodes the alpha-subunit of complex-V of the respiratory chain [21]. Progressive cerebellar ataxia is the primary feature of spinocerebellar ataxia type 7 (SCA7); moreover, pyramidal signs and eye movements abnormalities are observed. Retinal dystrophy (including cone-rod dystrophy and retinitis pigmentosa) is a frequent finding in SCA7 patients. SCA7 results from a dynamic mutation in the *ATXN7* gene, which is involved in chromatin remodeling [22]. Due to the symptoms of RP and neurological problems observed in P2, both NARP and SCA7 were suspected in the patient. NGS analysis allowed us to exclude these possibilities and confirm the clinical diagnosis of PCARP syndrome in P2. Retinitis pigmentosa and ataxia are also observed in abetalipoproteinemia (ABL) patients. This rare autosomal recessive disorder results from pathogenic variants in the *MTTP* gene, encoding microsomal triglyceride protein (MTP). ABL is characterized by low or absent plasma cholesterol levels, low-density lipoproteins (LDLs), and very-low-density lipoproteins (VLDLs). Additionally, patients have fat malabsorption and acanthocyte red blood cells [23]. In the congenital disorder of glycosylation, type 1a (PMM2-CDG) patients’ phenotype presentation is dominated by neurologic abnormalities such as psychomotor disability, seizures, hypotonia, and ataxia. Moreover, symptoms of retinitis pigmentosa are observed. PMM2-CDG results from defects in glycoprotein biosynthesis. The gene involved in disease pathogenesis is *PMM2*, which encodes essential for N-glycosylation enzyme phosphomannomutase 2 [24]. 

All patients presented in this study were diagnosed with distinct, ultra-rare genetic disorders. Nevertheless, they all show great phenotype variability in both ophthalmological and neurological manifestations, which resulted in misdiagnosis or delayed diagnosis in the past. Patients underwent several tests to make a clinical diagnosis, which shows the importance of a multidisciplinary approach in patients with such complex phenotypes. In recent years, with the improvement of NGS methods, the ratio of identified pathogenic gene variants has increased, thus giving, in many cases, the opportunity to finally confirm the clinical diagnosis, especially in ultra-rare genetic disorders with overlapping phenotypic features. 

## 4. Materials and Methods

### 4.1. Clinical Examination

Ophthalmologic examinations included measurement of central visual acuity, eye fundus examination, electroretinography (ERG) (P1 and P3), and optical coherence tomography (OCT). An audiological study performed in P1 and P2 included pure tone audiometry analysis (PTA). The neurological examination included magnetic resonance imaging (MRI) of the head. An electromyography examination was conducted in P2. Additionally, in P2, the phytanic acid and vitamin E serum levels were measured. Tandem mass spectrometry (tandem-MS) and gas chromatography–mass spectrometry (GC–MS) analyses were performed in P3. 

Written informed consent was obtained from the patients and family members (only those available for testing) prior to molecular analyses. All procedures were performed under the ethical standards of the institutional and national research committee and with the 1964 Helsinki Declaration and its later amendments or comparable ethical standards.

### 4.2. Molecular Analyses

Blood samples were collected from all patients. Genomic DNA was extracted from peripheral blood leukocytes by using the MagCore^®^ HF16 Automated Nucleic Acid Extractor and Genomic DNA Large Volume Whole Blood Kit (RBC Bioscience Corp., New Taipei, Taiwan) and from the buccal cells (patient 1 and patient 3 parents for segregation analysis) according to a kit protocol (Nucleo-Spin Tissue; Machery-Nagel, Düren, Germany).

#### 4.2.1. Patient 1 (P1)

The entire coding region of the *ABCA4* gene was sequenced (Asper Biogene, Tartu, Estonia). Multiplex ligation-dependent probe amplification (MLPA) analysis of the *PMP22* gene according to the standard protocol was performed (SALSA MLPA Probemix P033 CMT1, MRC Holland, Amsterdam, The Netherlands). The analysis included all exons (1–5) of the *PMP22* gene. NGS panel associated with Charcot–Marie–Tooth neuropathy (Charcot–Marie–Tooth Neuropathy Panel, version 1, March, 2016) encompassing 86 genes (Appendix A) was made (Blueprint Genetics, Espoo, Finland). Sequencing was conducted on an Illumina NextSeq (Illumina, San Diego, CA, USA). The human reference genome (hg19) was used. Mitochondrial DNA sequencing (Genomed, Warszawa, Poland) was carried out on an Illumina MiSeq (Illumina, San Diego, CA, USA). Variant pathogenicity was assessed using the MITOMAP database (https://www.mitomap.org/MITOMAP, accessed on 10 January 2023). Finally, the DNA sample was subjected to exome capture and high-throughput sequencing (WES). Target enrichment was conducted using SureSelect Human Exome V7 (Agilent Technologies, Santa Clara, CA, USA), and 150 bp paired-end reads sequencing was carried out on Illumina NovaSeq 6000 (Illumina, San Diego, CA, USA). A variant-discovery pipeline was built based on the GATK Best Practices. The human reference genome (hg38) was used. Filtering and interpretation of NGS data were performed using Ensembl/Variant Effect Predictor [25] and Exomiser [26] running the hiPHIVE algorithm. Frequent (AF > 0.001 in the population bases) and benign variants (according to ClinVar) were discarded at the filtering stage. Detected variants were referred to online databases of genomic variants, including HGMD^®^Professional, gnomAD SVs (v2.1) (genome Aggregation Database), ClinVar, and the Leiden Open Variation Database (LOVD). Copy-number variants were called using GATK gCNV software (v4.1.0.0). Quantitative real-time PCR (qPCR) was used to validate the CNV result in family 1. qPCR and Sanger sequencing were used to narrow down the deletion’s genomic coordinates in P1 (Appendix A). SYBR Green PCR Master Mix (Applied Biosystems, Waltham, MA, USA) was used to perform the qPCR reaction. All reactions were run in in triplicate on a ViiA™ 7 thermal cycler (Applied Biosystems). ALB and F8 were used for normalization; the F8 was the internal sample control (based on sex determination). The comparative 2−ΔΔCT method was used for the analysis, with noncommercial healthy control DNA as a calibrator. The thermal profile included the initial denaturation (95 °C for 10 min) followed by 40 quantification cycles (95 °C for 30 s and 60 °C for 60 s) and the melting curve cycle (95 °C, 15 s; 60 °C, 60 s and 95 °C, 15 s). For the breakpoint sequencing, the PCR conditions were as follows: initial denaturation step at 96 °C for 3 min followed by 20 cycles of denaturation at 94 °C for 30 s, annealing at 65 °C for 30 s, with temperature starting from 65 °C, decreasing to 55 °C (touchdown PCR −0.5 °C per cycle, 30 cycles with annealing temperature 55 °C), elongation at 72 °C for 1 min, with final elongation at 72 °C for 10 min. The obtained PCR products were sequenced. For primer sequences for qPCR and their genomic coordinates, see Appendix A. A schematic representation of the location of qPCR primers used to confirm the deletion and to narrow down the sequence with the breakpoints is presented in Appendix A.

#### 4.2.2. Patient 2 (P2)

MtDNA sequencing was performed in the patient (details are described above in P1). Next, an analysis of a number of CAG repeats in the genes: *ATXN1*, *ATXN2*, *ATXN3*, and *ATXN7* using PCR followed by capillary electrophoresis was conducted. Finally, WES was carried out. The details are described in P1. A confirmation study was performed by applying PCR followed by Sanger sequencing. 

#### 4.2.3. Patient 3 (P3)

A diagnostic NGS panel associated with retinal disorders, encompassing 317 genes (Appendix A), was performed in the patient (Genomed, Warszawa, Poland). Sequencing was conducted on an Illumina NextSeq 500 (Illumina, San Diego, CA, USA). The human reference genome (hg38) was used. The analysis details are described above. Sanger sequencing was performed to validate the variants and confirm their biparental origin. For details, see the Appendix A. 

Specific primers for all amplifications were designed using the online Primer3Plus tool (version: 3.2.6). For the detailed list of primers, see Appendix A. The final pathogenicity of detected variants in all patients was analyzed in line with the American College of Medical Genetics (ACMG) classification [27]. Variants identified in the studied probands, as well as variants previously described in genes *ABHD12, FLVCR1*, and *PNPLA6*, were shown in Appendix A.

## Figures and Tables

**Figure 1 ijms-25-05759-f001:**
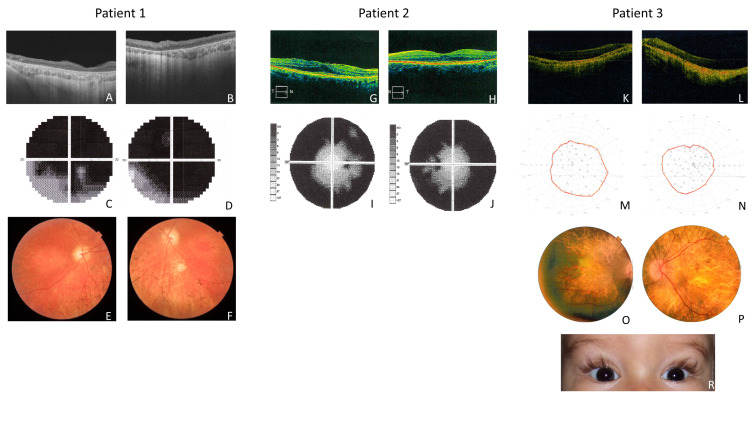
Patients’ ophthalmological examination results. **Patient 1** (**A**,**B**)—Optical coherence tomography (OCT) showed RPE and photoreceptors atrophy with retinal thinning; (**C**,**D**)—Loss of the central visual field in both eyes; (**E**,**F**)—Fundus photography showed atrophy of the optic nerves, attenuated retinal vessels, and bone spicules. **Patient 2** (**G**,**H**)—OCT showed peripheral RPE and photoreceptors atrophy with retinal thinning; (**I**,**J**)—Concentric narrowing (to approximately 20–30 degrees) of the visual field in both eyes. **Patient 3** (**K**,**L**)—OCT showed generalized atrophy of the RPE and photoreceptors; (**M**,**N**)—Concentric narrowing (to approximately 40–70 degrees) of the visual field in both eyes; (**O**,**P**)—Fundus photography showed atrophy of the optic nerves and atrophic macula with pigment clumping; (**R**)—Clinical photography of the patient, showing trichomegaly of eyelashes.

**Figure 2 ijms-25-05759-f002:**
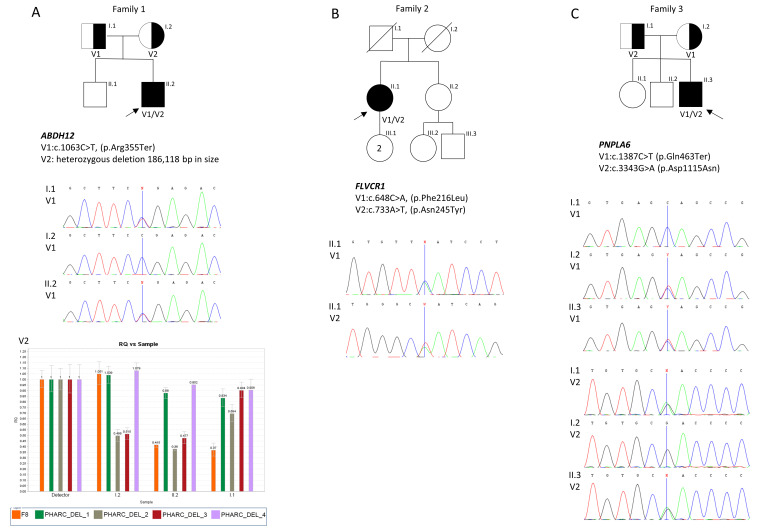
Pedigrees and segregation of the variants associated with the disease in patients’ families ((**A**)—family 1, (**B**)—family 2, (**C**)—family 3). Identified variants are placed below the pedigree. Causal variants are indicated with V1 and V2, and + indicates a wild-type allele. Arrows point to probands. Chromatograms and qPCR results show variants identified in patients and their parents. qPCR results in family 1 show a lower copy number in the affected proband (II.2) and his mother (I.2), confirming the described deletion (V2). The unaffected father (I.1) was negative for the lesion. Error bars represent standard deviation. Normalization was performed against the F8 gene located on the X chromosome. PHARC_DEL_1 and PHARC_DEL_4 primers were located in the regions flanking the deletion. PHARC_DEL_2 and PHARC_DEL_3 primers comprised the deleted sequence (Appendix A).

**Table 1 ijms-25-05759-t001:** Clinical symptoms of the patients with retinitis pigmentosa and ataxia.

Patient ID/Family	Current Age/ Gender	Ophthalmic Symptoms								Polyneuropathy/Cerebellar Ataxia	
		Night Blindness/Age	Anterior Segment	Arteriolar Attenuation	Macula	Peripheral Retina	BCVA RE/LE	VF Restriction	ERG Results	Ataxia	Polyneuropathy
P1/F1	33/M	+/13	-	-	Mild atrophic degenerative changes	Decreased RPE pigmentation	0.2/0.4	Mild central scotoma	From decreased (70%, at 19 years) to extinguished (at 30 years)	+	Sensorimotor
P2/F2	61/F	+/20	Cataract	+	ERM	Bone-spicules	0.1/0.4	Concentric narrowing	ND	+	Severe sensorineural
P3/F3	15/M	+/2	-	+	Maculopathy	Bone-spicules	0.1/0.1	Concentric narrowing	Extinguished	+	-

BCVA—best-corrected visual acuity. RE—right eye. LE—left eye. M—male. F—female. VF—visual field. ERM—epiretinal membrane. ND—no data.

**Table 2 ijms-25-05759-t002:** DNA variants identified in patients by NGS.

Patient/ Family	Gene	Transcript	Variant Classification		Pathogenicity Prediction in Protein Level			ACMG Classification	Allele Frequency (gnomAD)	Molecular Method of Searching the Variants
			Nucleotide	Protein	SIFT	PolyPhen-2	CADD			
P1/F1	*ABHD12*	NM_001042472.3	c.1063C˃T (rs200536497) **NC_000020.11:g[(25317079_25319906)_ (25505369_?)del];[=]**	p.Arg355Ter -	- -	- -	D -	Pathogenic -	0.0000192 -	WES
P2/F2	*FLVCR1*	NM_014053.4	**c.648C˃A** **c.733A˃T**	**p.Phe216Leu** **p.Asn245Tyr**	D D	LD U	U D	Likely pathogenic Likely pathogenic	- -	WES
P3/F3	*PNPLA6*	NM_001166114.2	**c.1387C˃T** c.3343G˃A (rs372763461)	**p.Gln463Ter** p.Asp1115Asn	- D	- LD	D D	Likely pathogenic Likely pathogenic	- 0.0000093	NGS panel

WES—whole-exome sequencing, D—damaging, LD—likely damaging, U—uncertain. American College of Medical Genetics (ACMG) classification was obtained through the Varsome online available tool. New variants identified in this study are bolded.

## Data Availability

The data used in this study are available from the corresponding author upon request.

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
