# Peer review of "Coexistence of Retinitis Pigmentosa and Ataxia in Patients with PHARC, PCARP, and Oliver–McFarlane Syndromes"

_ijms, 2024, doi:10.3390/ijms25115759_

Round 1

Reviewer 1 Report

Comments and Suggestions for Authors

The authors present a case report investigating the genomic causes of patients with PHARC, PCARP, and Oliver-McFarlane syndromes. Overall, this manuscript is well-written. Here are some suggestions for the authors' consideration:

1.    Patient 1's heterozygous mutation was a 186,118 bp (chr20:25,319,102-25,505,220) deletion, not kb.

2.    The qPCR results from Figure 2 were difficult to read (possibly due to the low resolution of the downloaded PDF). The authors should provide a detailed description of the figure in the Results section or the Figure legend, including the significance of each primer, the region of the genome it covers, the specific part of the genes targeted, and the interpretation of the results. Additionally, are the error bars shown in the bar plot attributed to technical repeats of the qPCR? The authors should clarify what these error bars represent.

3.    For known variants, the authors should provide the minor allele frequency.

4.    The descriptions of the qPCR and breakpoint methods should be included in the main manuscript. I would suggest the authors provide a diagram illustrating the target genes and location of all the PHARC_DEL primers.

5.    The authors could provide a more detailed description of variant calling from WEG/WGS in the Methods section.

Reviewer 2 Report

Comments and Suggestions for Authors

The authors have shown the clinical and genetic analysis of three patients with retinitis pigmentosa coexisting with cerebellar ataxia in this case reports. The authors used the whole exome sequencing and targeted NGS and reported four novel variants in the following genes: ABHD12, FLVCR1, and PNPLA6. The study is interesting and novel. The identification of novel disease variants is important for the research community. My comments are provided below

1. The authors should make a diagram of the variants in their respective gene.

2. The authors didnot show any any data on the effect of these novel variants on the gene expression level. The QRTPCR data is also missing the statistical analysis.

3. The authors didnot discuss the limitation of the study. The sequencing methods are error prone and the low numbers of patients in the case reports with missing control should be discussed.

4. The expression level of target genes in protein level is missing.

Minor comments

1. The label for subpanels in figure1 arenot clear, and it is also missing in Figure 2.

Comments on the Quality of English Language

The english language needs to improved. The sentences are very long. The hyphen in 'con-firmed' should be removed.

Round 2

Reviewer 2 Report

Comments and Suggestions for Authors

The authors have addressed all my concerns in the revised manuscript. I support the publication of the manuscript.

Comments on the Quality of English Language

There are minor spelling error in the text.